# Resurgence of Chloramphenicol Resistance in Methicillin-Resistant *Staphylococcus aureus* Due to the Acquisition of a Variant Florfenicol Exporter (*fexAv*)-Mediated Chloramphenicol Resistance in Kuwait Hospitals

**DOI:** 10.3390/antibiotics10101250

**Published:** 2021-10-15

**Authors:** Edet E. Udo, Samar S. Boswihi, Bindu Mathew, Bobby Noronha, Tina Verghese

**Affiliations:** Department of Microbiology, Faculty of Medicine, Kuwait University, Kuwait City 12037, Kuwait; samar.boswihi@ku.edu.kw (S.S.B.); bindujmk@gmail.com (B.M.); bobby.vaz@gmail.com (B.N.); tinajay@gmaail.com (T.V.)

**Keywords:** chloramphenicol resistance, florfenicol exporter, MRSA, antibiotic resistance, molecular typing

## Abstract

Following a surge in the prevalence of chloramphenicol-resistant methicillin-resistant *Staphylococcus aureus* (MRSA) in Kuwait hospitals, this study investigated the genotypes and antibiotic resistance of the chloramphenicol-resistant isolates to ascertain whether they represented new or a resurgence of sporadic endemic clones. Fifty-four chloramphenicol-resistant MRSA isolates obtained in 2014–2015 were investigated. Antibiotic resistance was tested by disk diffusion and MIC determination. Molecular typing was performed using *spa* typing, multilocus sequence typing, and DNA microarray. Curing and transfer experiments were used to determine the genetic location of resistance determinants. All 54 isolates were resistant to chloramphenicol (MIC: 32–56 mg/L) but susceptible to florfenicol. Two chloramphenicol-resistance determinants, florfenicol exporter (*fexA*) and chloramphenicol acetyl transferase (*cat*), were detected. The *fexA*-positive isolates belonged to CC5-ST627-VI-t688/t450/t954 (*n* = 45), CC5-ST5-V-t688 (*n* = 6), whereas the *cat*-positives isolates were CC8-ST239-III-t037/t860 (*n* = 3). While *cat* was carried on 3.5–4.4 kb plasmids, the location of *fexA* could not be established. DNA sequencing of *fexA* revealed 100% sequence similarity to a previously reported *fexA* variant that confers chloramphenicol but not florfenicol resistance. The resurgence of chloramphenicol resistance was due to the introduction and spread of closely related *fexA*-positive CC5-ST5-V and CC5-ST627-VI clones.

## 1. Introduction

Chloramphenicol is a broad-spectrum antibiotic that was derived from *Streptomyces venezuelae* initially [1,2,3] but has also been produced synthetically [3,4]. Chloramphenicol was introduced into clinical practice in 1940s for the treatment of infections caused by Gram-positive and Gram-negative organisms such as *Staphylococcus aureus*, *Streptococcus pneumoniae*, *Salmonella Typhi*, *Haemophilus influenzae*, *Escherichia coli*, and *Neisseria meningitides* [1,5].

Chloramphenicol is bacteriostatic but may exert bactericidal activity at higher concentrations [2]. Chloramphenicol diffuses through the bacterial cell wall and binds to the bacterial 50S ribosomal subunit. The binding interferes with peptidyl transferase activity and prevents the transfer of amino acids to the growing peptide chains and blocks peptide bond formation resulting in the blocking of bacterial protein synthesis [4,6,7].

Whereas chloramphenicol and some derivatives such as thiamphenicol have been used in human medicine over the years for treating bacterial infections due to its effectiveness, low cost, and broad spectrum of activity, florfenicol, a fluorinated derivative of chloramphenicol, is licensed for veterinary use and is used exclusively in veterinary medicine [4,5,8].

Resistance to chloramphenicol emerged following the clinical and extensive use of chloramphenicol in human and veterinary medicine [5,9]. The most common mechanism of chloramphenicol resistance in *Staphylococcus aureus* is enzymatic inactivation by chloramphenicol acetyltransferase (CAT) [4]. The other mechanisms of chloramphenicol resistance are the presence of an efflux mechanism due to chloramphenicol/florfenicol exporter (fexA) [4] and the 23S rRNA methyl transferase (*cfr*) that also mediate resistance to linezolid [4].

Chloramphenicol acetyltransferase inactivates chloramphenicol and thiamphenicol but not florfenicol, and consequently, chloramphenicol-resistant strains, in which resistance is exclusively mediated by CAT, are susceptible to florfenicol [4].

The prevalence of chloramphenicol resistance was high in methicillin-resistant *S. aureus* (MRSA) obtained from patients in Kuwait hospitals in the 1990s with a prevalence of 44% in 1996, but this gradually declined to 2.0% in 2004 [10]. The prevalence of chloramphenicol-resistant MRSA remained low at 2.0% until 2010 [11,12] and then increased again from 2.6% in 2011 to 9.6% in 2015 [12]. The chloramphenicol-resistant MRSA isolates reported in the 1990s to early 2000s belonged to the healthcare-associated MRSA clone ST239-MRSA-III [13] and harbored small 2.8–4.4 kb chloramphenicol-resistant plasmids [10].

The decline in the proportion of chloramphenicol resistance coincided with changes in the diversity of MRSA clones which witnessed the reduction in the proportion of ST239-MRSA-III and the increase in the number and types of community-associated MRSA clones [13]. Therefore, the observed increase in the proportion of chloramphenicol-resistant MRSA in Kuwait hospitals could be due to a resurgence of the ST239-MRSA-III clone that was prevalent in the 1990s or due to the introduction of a new chloramphenicol resistant MRSA clone into Kuwait public hospitals. Consequently, this study was conducted to investigate the genotypes of the chloramphenicol-resistant isolates obtained in 2014–2015 to ascertain whether the isolates represented the introduction of new clones or the resurgence of the previously endemic ST239-MRSA-III clones.

## 2. Results

### 2.1. Molecular Typing of Chloramphenicol MRSA Isolates

A total of 130 isolates constituting 7.8% of the MRSA isolates submitted for molecular typing in 2014–2015 were resistant to chloramphenicol (MIC ≥ 32 µg/mL). *Spa* typing of the 130 MRSA isolates revealed five *spa* types consisting of t688 (*n* = 125), t450 (*n* = 1), t954 (*n* = 1), t037 (*n* = 2) and t860 (*n* = 1). Fifty-four of the 130 isolates were then selected to reflect the different *spa* types, clinical samples, and hospital location and investigated further by MLST and DNA microarray to determine their clonal types and carriage of resistance and virulence genes.

The results, summarized in Table 1, showed that the isolates belonged to two clonal complexes (CC), CC5 and CC8, with the majority (51/54) belonging to CC5. The CC5 isolates were further subdivided into two groups based on the carriage of SCC*mec* types: SCC*mec* type V (*n* = 6) and SCC*mec* type VI (*n* = 45). The CC8 isolates carried SCC*mec* type III (*n* = 3).

The results of MLST performed on all 54 isolates revealed that the CC5 isolates belonged to ST5 (*n* = 6) and ST627 (*n* = 45), while the CC8 isolates belonged to ST239. The ST627 isolates harbored SCC*mec* type VI and were associated with *spa* types t688 (*n* = 42), t450 (*n* = 1) and t954 (*n* = 1), all the ST5 isolates harbored SCC*mec* type V and were associated with *spa* type t688, while the ST239 isolates harbored SCC*mec* type III and were associated with *spa* type t037 (*n* = 2) and t860 (*n* = 1).

The ST5 isolates belonged to a single strain type, ST5-MRSA-V-t688 (*n* = 6), whereas the ST627 isolates belonged to two strain types, ST627-MRSA-VI+SCCfus-t688/t450/(*n* = 43) and ST627-MRSA-VI-t688/t954 (*n* = 2).

### 2.2. Antibiotic Resistance Phenotypes and Genotypes

The isolates were susceptible to vancomycin (MIC ≤ 2 µg/mL), teicoplanin (MIC ≤ 2 µg/mL), linezolid, tigecycline, and rifampicin but were resistant to chloramphenicol (MIC 32–256 µg/mL) (*n* = 54), tetracycline (*n* = 48), trimethoprim (*n* = 45), fusidic acid (*n* = 45), erythromycin and clindamycin (*n* = 10), gentamicin and kanamycin (*n* = 3), ciprofloxacin (*n* = 3), and high-level mupirocin (*n* = 1) (Table 2).

All isolates harbored *blaZ*, which confers penicillin resistance. In addition, most of the isolates harbored *fexA* (*n* = 51), and three isolates were positive for *cat*, both of which confer chloramphenicol resistance. The tetracycline resistant isolates harbored *tet(K)* + *tet(M)* (*n* = 8), *tet(M)* (*n* = 45), or *tet(K)* (*n* = 1). The trimethoprim resistance gene *dfrS1* was detected in 45 isolates, and the fusidic acid resistance gene *fusC* was detected in 45 isolates. The erythromycin and clindamycin-resistant isolates carried *erm(C)* (*n* = 7) or *erm(A)* (*n* = 3). Isolates that were resistant to gentamicin and kanamycin were positive for *aacA-aphD* (*n* = 3), *aphA3* (*n* = 3) or *aadD* (*n* = 1). The single high-level mupirocin-resistant isolate was positive for *mupA* (Table 2).

### 2.3. Determination of Susceptibility to Florfenicol

As shown in Table 2, DNA microarray analysis detected the presence of *fexA*, which usually confers resistance to chloramphenicol and florfenicol, in 51 of the 54 chloramphenicol-resistant MRSA isolates. As susceptibility to florfenicol was not included in the initial disk susceptibility testing, additional testing of the isolates for florfenicol susceptibility was performed by determining the MIC of florfenicol. All isolates had florfenicol MIC of 4 µg/mL, indicating susceptibility to florfenicol. Florfenicol resistance is defined as florfenicol MIC ≥ 16 µg/mL [14].

### 2.4. Amplification of Chloramphenicol and Florfenicol Resistance Genes

Genomic DNA was obtained from the 54 MRSA isolates and used as templates in PCR assays against primers for *fexA*, which codes for chloramphenicol/florfenicol exporter; *cfr*, which codes for 23S rRNA methyltransferase, which confers resistance to phenicol, lincosamides, and oxazolidinones; and *cat*, which codes for chloramphenicol acetyltransferase. None of the isolates were positive for *cfr*. However, the 51 CC5 and 3 CC8 isolates were positive for *fexA* and *cat*, respectively. The *fexA*-positive isolates yielded the expected PCR products of 1272 bp, while the cat-positive isolates yielded products of 748 kb (Figure 1).

### 2.5. DNA Sequencing of fexA

The *fexA*-positive PCR products of two isolates, selected to represent the two CC5 genetic backgrounds, ST5-MRSA-V-t688 and ST627-MRSA-VI-t688, were sequenced to determine their relatedness. The nucleotide sequences obtained with *fexA* from both isolates were identical to each other, and when compared with f*exA* sequences available in the GenBank, the result revealed a 100% similarity to *fexA* sequences found in GenBank with accession number KX2300476 [15]. Analysis of the sequences revealed mutations leading to amino acid substitutions isoleucine 131-Valine (Ile131Val) and Proline 321 to Threonine (Pro321Thr).

### 2.6. Virulence Genes of Chloramphenicol Resistant Isolates

DNA microarray analysis revealed that the isolates were positive for a range of virulence determinants, including regulatory genes, enterotoxins, leucocidins, hemolysins, immune evasive clusters, adhesion factors, clumping factors, biofilm-associated genes, proteases, and restriction modification systems. A list of some important virulence genes is presented in Appendix A. All isolates were positive for the regulatory genes, *sarA* (staphylococcal accessory regulator A) and *saes* (histidine protein kinase), leukocidins and biofilm-associated genes, clumping factors A and B (*clfA*, *clfB*), fibronectin binding protein (*fib*), fibronectin binding proteins A and B (*fnbA*, f*nbB*), and major histocompatibility complex class II (*Map*). All isolates were positive for genes coding for hemolysins, except one ST239-MRSA-III isolate that was negative for genes for hemolysin alpha (*hla*). The isolates were all negative for genes that code for Panton valentine leukocidin (*lukF-PV-lukS-PV*), toxic shock syndrome toxin (*tst1*), enterotoxins, *seb*, *sec*, and *sel*, exfoliative toxins, epidermal cell differentiation inhibitors, and arginine catabolic mobile elements (ACME). However, whereas the CC5 isolates were positive for *agrII* (accessory gene regulator II), the ST239-MRSA isolates were positive for *agr I* (accessory gene regulator I). Similarly, whereas the CC5 isolates were positive for capsular polysaccharide type 5 (*cap5*), the ST239-MRSA isolates were positive for capsular polysaccharide type 8 (c*ap8*). The isolates also differed in the carriage of genes for enterotoxins, collagen binding adhesin (*cna*) serine protease E (*splE*) and Type I restriction modification system (Appendix A).

### 2.7. Genetic Location of Chloramphenicol/Florfenicol Resistance

Plasmid analysis of the 54 isolates revealed the presence of one to four plasmids in a cell that ranged in size from < 2.0 kb to *c*40 kb (Figure 2). Most of the isolates harbored a single plasmid of c.28 kb. To determine the association of the plasmids with resistance phenotypes, seven isolates representing different plasmid profiles were selected for curing and transfer experiments.

#### 2.7.1. Curing of Resistance and Plasmids

For curing experiments conducted on each isolate, a maximum of 960 colonies were screened for resistance following growth at 43 °C. Table 3 summarizes the plasmids and resistance lost following curing.

There was no loss of chloramphenicol resistance in any of the 960 colonies screened for all selected isolates, including isolate #14071 (ST5-MRSA-V-t688) carrying two plasmids of c.40 kb and c.28 kb and the isolates harboring single c.28 kb plasmids (CC5-MRSA isolates). Resistance to chloramphenicol, gentamicin and kanamycin, and high-level mupirocin were lost from isolate #14287 (ST239-III-t037). Isolate #14287 contained two plasmids of 40 kb and 4.4 kb. The loss of chloramphenicol resistance was accompanied by loss of a 4.4 kb plasmid and the loss of gentamicin and kanamycin, and mupirocin resistance was accompanied by loss of c.40 kb. Loss of chloramphenicol from isolates #14299 (ST239-MRSA-III-t037) and #14387 (ST239-MRSA-III-t860) was accompanied by loss of 3.5 kb plasmids. Resistance to erythromycin and clindamycin were lost accompanied by loss of 2.8 kb plasmids in two isolates, #13973 (ST5-MRSA-V-t688) and #14387 (ST239-MRSA-III-t860).

#### 2.7.2. Transfer of Resistance and Plasmids

Experiments were conducted to transfer plasmids from the representative isolates and associate the resistance phenotypes with plasmids using conjugation, mobilization, and mixed culture transfer experiments. The results of the transfer experiments are summarized in Table 4.

Only isolate #14284 (ST239-MRSA-III-t860) transferred resistance to gentamicin, high-level mupirocin, and chloramphenicol in conjugation experiments. Transconjugants on gentamicin and mupirocin selection plates were resistant to both gentamicin and mupirocin and contained a c.40.0 kb plasmid, indicating that both resistance determinants were co-located on the same plasmid. Transconjugants on chloramphenicol selection were resistant to chloramphenicol or chloramphenicol and mupirocin and gentamicin. Colonies resistant to chloramphenicol carried a 4.4 kb plasmid whereas colonies resistant to chloramphenicol, mupirocin, and gentamicin carried two plasmids of 40 kb and 4.4 kb in size. These results indicate that the 40 kb plasmid carrying resistance to gentamicin and mupirocin was conjugative and was mobilizing the 4.4 kb chloramphenicol resistance plasmid (Figure 3).

As the rest of the isolates failed to transfer resistance in direct conjugation, mobilization using conjugative plasmids pWBG626 (Gm^R^) and pXU12 (mup^R^) was performed. Both conjugative plasmids mobilized chloramphenicol resistance accompanied by the transfer of 3.5 kb plasmids in isolates #14299 and #14387, and erythromycin resistance accompanied by the transfer of a 2.8 kb plasmid in isolates #13973 and #14387. Chloramphenicol resistance could not be transferred by conjugation or mobilization from isolates #14098, #14314, and #14434, carrying only 28.0 kb plasmids. None of the isolates transferred chloramphenicol resistance in the MCT experiments.

### 2.8. Amplification of Plasmid Borne Chloramphenicol Resistance Genes

The 4.4 kb and 3.5 kb chloramphenicol resistance plasmids were used as templates in PCR experiments using the *cat*, *cfr*, and *fexA* primers to confirm the carriage of these genes on the plasmids. The 3.5 kb and 4.4 kb plasmids yielded positive results only for *cat*. The *cfr* and *fexA* were not amplified.

## 3. Discussion

As the epidemiology of antibiotic-resistant *S. aureus* is constantly changing, the emergence of antibiotic-resistant pathogens into a healthcare facility can be due to the use of concerned antibiotics to treat infections in the facility [16,17] or the introduction of a previously resistant pathogen into the facility independent of antibiotic use [18,19,20]. The epidemiology of MRSA in Kuwait hospitals has changed substantially since the 1990s with the ST239-MRSA-III clone that was dominant in the 1990s gradually replaced by diverse CA-MRSA clones starting in the early 2000s [13,21,22]. One of these changes was the increase in the proportion of chloramphenicol-resistant MRSA isolates, from 2.6% in 2011 to 9.6% in 2015 [12]. The resurgence in the prevalence of chloramphenicol resistance in Kuwait MRSA isolates could not be explained by increased chloramphenicol use, because chloramphenicol is hardly used for treatment in Kuwait public hospitals. This study was therefore initiated to explain the possible causes of the resurgence in chloramphenicol-resistant MRSA in Kuwait hospitals.

Molecular typing revealed that most of the chloramphenicol-resistant isolates belonged to CC5-ST627-VI-t688/t450/t954 (45/54) or CC5-ST5-V-t688 (6/54), while a small number (3/54) belonged to ST239-III/t037/t860. This clearly indicates that the surge in the proportion of chloramphenicol-resistant MRSA isolates was due to the introduction and spread of closely related novel MRSA clones belonging to ST627-VI-t688 and ST5-V-t688. The ST239-III-t037/t860 that resembled the chloramphenicol-resistant MRSA strains that were dominant in the 1990s [13] were isolated sporadically in this study and were not responsible for the increase in the proportion of chloramphenicol resistant isolates observed in this study.

The chloramphenicol-resistant MRSA isolates harbored genes for a variety of virulence factors. Isolates of both clonal complexes shared genes for aureolysin, leukocidins, hemolysins, biofilm formation, clumping factors A and B, and fibronectin binding proteins A and B. However, the isolates differed in the carriage of gene for enterotoxins, collagen binding adhesin, accessory gene regulators, and capsular polysaccharide. Whereas the CC5-MRSA isolates were positive for accessory gene regulator II (*agr*
*II*) and capsular polysaccharide type 5 (*cap5*), the ST239-MRSA isolates were positive for *agr*
*I* and *cap8*. In addition, whereas the CC5-MRSA isolates lacked the enterotoxin genes, *sek and seq*, the ST239-MRSA isolates were positive for them. These characteristics are consistent with the carriage of virulence determinants in these clonal complexes [15,23].

Our records show that a single isolate of CC5-MRSA-VI+SCCfus-t688, and eight isolates of CC5-MRSA-V-t688 were detected for the first time in Kuwait in 2010 [13]. The results of this study suggest that the CC5-MRSA-VI+SCCfus clone, carrying the unique composite genetic element consisting of SCC*mec* VI and fusidic acid resistance determinant, *fusC*, which was isolated from a single patient in 2010, has successfully spread to become the dominant chloramphenicol-resistant clone in Kuwait hospitals. Whereas the CC5-ST627-VI+SCCfus-t688/t450/t954 is an emerging lineage in Kuwait, the ST5-V-t688 isolates were previously reported in Western Australia, where it was designated as WA-MRSA-11/34/35/ [23,24], Ireland [25], and Turkey [26].

This study revealed that two chloramphenicol resistance determinants, namely *cat*, which codes for chloramphenicol acetyltransferase, and *fexA*, which codes for chloramphenicol/florfenicol exporter [27,28], conferred chloramphenicol resistance in Kuwait hospitals in the study period. Whereas *cat* was detected in the ST239-III-t037/t860 isolates, *fexA* was present in the ST627-VI-688/t450/t954 and ST5-V-t688 isolates. These observations confirmed that the resurgence of chloramphenicol-resistant MRSA in Kuwait hospitals was due to the acquisition of novel MRSA clones carrying *fexA*-mediated chloramphenicol resistance.

The chloramphenicol acetyltransferase (cat) determinants in *S. aureus* of human and animal origin are usually located on small multi-copy plasmids of 2.9–5.1 kb, including pC221, pC223, pC194, pSC20, and pSC23 [4,9,10,29,30]. Similarly, curing and transfer experiments identified the *cat* determinant in the ST239-III-t037/t860 isolates on small 3.5–4.4 kb plasmids in this study. In contrast, the genetic location of *fexA* could not be determined in this study because it could neither be lost on curing nor transferred in conjugation or mixed-culture transfer experiments. The failure to cure or transfer *fexA* in these isolates suggests that it may be located on the bacterial chromosome. However, a plasmid location cannot be discounted since the isolates harbor plasmids whose phenotypes have not been determined. In addition, *fexA* has been reported on c.33–35 kb plasmids, on the chromosomal DNA, and as part of a transposon designated Tn558 [8,31]. Therefore, further studies are required to determine the genetic location of *fexA* in these isolates.

Surprisingly, although *fexA* usually confers combined resistance to chloramphenicol and florfenicol in *S. aureus* obtained from animals [4,32,33,34] and humans [35], the *fexA*-positive isolates in this study were resistant to chloramphenicol (MIC 32–256 µg/mL) but susceptible to florfenicol (MIC 4 µg/mL). However, literature searches revealed a previous report of a *fexA* variant (fexAv) that confers chloramphenicol but not florfenicol resistance that was present in MRSA isolated from chicken meat [15]. Another *fexA* variant was also detected in *S. pseudintermedius* [36]. DNA sequence analysis of the *fexA* variant in the MRSA strain W333 that was obtained from chicken meat revealed mutations leading to amino acid substitutions in isoleucine 131-Valine (Ile131Val) and Proline 321 to Threonine (Pro321Thr) [15]. A comparison of the DNA sequence of the *fexA* in the isolates in this study yielded 100% similarity to the variant form of f*exA* (fexAv) reported in MRSA W333, including mutations leading to amino acid substitutions in isoleucine 131-Valine (Ile131Val) and Proline 321 to Threonine (Pro321Thr), confirming the presence of the *fexA* variant (f*exAv*) in our isolates. A related *fexA* variant exhibiting amino acid substitutions Gly33Ala and Ala37Val was reported in *Staphylococcus pseudinermedius* [36].

Besides the similarities in the DNA sequence of *fexAv* in our isolates and in W333, the isolates belonged to the same *spa* type, t688, and were resistant to penicillin G and tetracycline mediated by *blaZ*, *tet(K)*, or *tet(M)* suggesting relatedness between the isolates. Although no epidemiological relationship could be established between our isolates obtained from human patients and W333 that was cultured from chicken meat that originated in Egypt [15], there is a large population of expatriate workers from Egypt in Kuwait. In addition, published studies show that MRSA strains similar to W333 are common in Egypt. For example, *fexA*-positive ST5-VI-t688 MRSA isolates constituted 6% of MRSA isolates obtained from human patients in a hospital in Alexandria, Egypt [37], and *fexA*-positive CC5-MRSA-V/VT were recovered from milk of cattle and buffaloes with mastitis also in Egypt [38,39] indicating the widespread distribution of *fexA* in MRSA in diverse hosts in the country. The presence of the *fexA*-positive clone in cattle and buffaloes with subclinical mastitis poses potential risk to humans since the consumption of milk obtained from cattle or buffaloes with subclinical mastitis can be a potential source of transmission of resistant *S. aureus* to humans [38,39]. Interestingly, Antonelli et al., [40] detected a novel gene, *poxtA*, that codes for reduced susceptibility to phenicols-oxazolidinone-tetracycline in a human MRSA isolate. As *poxtA* was initially reported in *Enterococcus faecalis*, *Enterococcus faecium*, and *Pediococcus acidilactici* of animal origin, its detection in a human MRSA isolate suggests that it was acquired from an animal pathogen. Similarly, the *fexAv* reported in this study was also probably acquired from an animal source. These studies highlight the increasing transmission of antibiotic resistance determinants from animals to humans.

In conclusion, this study revealed that the resurgence in chloramphenicol resistance in MRSA obtained in Kuwait hospital was due to the introduction of a new chloramphenicol-resistant MRSA clone harboring a variant *fexA* that mediated resistance to chloramphenicol but not florfenicol. The variant *fexA* shared 100% sequence similarity with a *fexA* variant that was detected in MRSA isolated from chicken meat. The similarity of these isolates to those obtained from chicken meat and milk from cattle and buffalos points to the ongoing intrusion of MRSA isolates from livestock to humans. The t688-CC5-MRSA-V/VI isolates has continued to spread and constituted 9% of the MRSA that were isolated from patients in Kuwait hospitals in 2020 (unpublished report).

## 4. Materials and Methods

### 4.1. Bacterial Strains

The 54 chloramphenicol-resistant MRSA isolates were cultured from patient samples in nine hospitals in Kuwait as part of routine diagnostic microbiological investigations from 1 May 2014 to 31 December 2015. The initial identification of the isolates was performed in the diagnostic laboratories, and the MRSA isolates were later submitted for molecular typing at the MRSA Reference Laboratory, located at the department of Microbiology, Faculty of Medicine, Kuwait University, Kuwait. Isolates were obtained from the nose (*n* = 13), skin and soft tissues (*n* = 14), high vaginal swabs (*n* = 8), blood (*n* = 4), groin (*n* = 5), and miscellaneous sources (*n* = 10).

### 4.2. Antibiotic Susceptibility Testing

Susceptibility testing was performed by the disk diffusion method [41] for the following antibiotics: benzyl penicillin (2U), cefoxitin (30 µg), kanamycin (30 µg), mupirocin (200 µg), gentamicin (10 µg), erythromycin (15 µg), clindamycin (2 µg), chloramphenicol (30 µg), tetracycline (10 µg), tigecycline (15 µg), trimethoprim (2.5 µg), fusidic acid (10 µg), rifampicin (5 µg), ciprofloxacin (5 µg), and linezolid (30 µg). Methicillin resistance was confirmed by detecting PBP 2a using a rapid latex agglutination kit (Denka-Seiken, Tokyo, Japan) according to the manufacturer’s instruction [42].

#### Determination of the Minimum Inhibitory Concentration (MIC)

The MIC of cefoxitin, vancomycin, teicoplanin and chloramphenicol were determined with Etest strips (BioMerieux, Marcy l’ Etoile, France) following the manufacturer’s instructions. The MIC of florfenicol was determined by agar dilution method using florfenicol powder (Sigma-Aldrich, St. Louis, MI, USA) in Mueller Hinton Agar plates with dilutions ranging from 0.5 µg/mL to 128 µg/mL. *S. aureus* strain ATCC 29,213 was used as quality control strain for susceptibility testing. The results were interpreted according to CLSI standard VET01 [14].

### 4.3. Molecular Typing of Isolates

#### 4.3.1. DNA Isolation for Amplification

DNA isolation was performed as described previously [43]. Three to five identical colonies of an overnight culture were picked using a sterile loop and suspended in a microfuge tube containing 50 µL of lysostaphin (150 µg/mL) and 10 µL of RNase (10 µg/mL) solution. The tube was incubated at 37 °C in the heating block (Thermo Mixer, Eppendorf, Hamburg, Germany) for 20 min. To each sample, 50 µL of proteinase K (20 mg/mL) and 150 µL of Tris buffer (0.1 M) were added and mixed by pipetting. The tube was then incubated at 60 °C in the water bath (VWR Scientific Co., Shellware Lab, United States) for 10 min. The tube was transferred to a heating block at 95 °C for 10 min to inactivate proteinase K activity. Finally, the tube was centrifuged, and the supernatant containing extracted DNA was stored at 4 °C till used for PCR.

#### 4.3.2. Amplification of Chloramphenicol Resistance Genes

The isolates were investigated for the presence of genes encoding chloramphenicol acetyl transferase (*cat)*, florfenicol-chloramphenicol exporter (*fexA*), and 23S rRNA methyltransferase (*cfr)* that codes for resistance to phenicol, lincosamides, oxazolidinone, pleuromutilin, and streptogramin A. For the amplification of *cat*, the primers cat-F- 5′- GCG AAC GAA AAA CAA TTG CA -3′ and cat-R- 5′- TGA AGC TGT AAG GCA ACT GG-3′, published previously by Kim et al. [44], were used. For *fexA*, the following primers were used: fexA-F 5′-GTA CTT GTA GGT GCA ATT ACG GCT GA -3′ and fexA-R 5′-CGC ATC TGA GTA GGA CAT AGC GTC-3′. For *cfr*, the primer pair used was: cfr-F 5′- TGA AGT ATA AAG CAG GTT GGG AGT CA-3′ and cfr-R 5′-ACC ATA TAA TTG ACC ACA AGC AGC-3′ [8]. The amplified products were examined by agarose gel electrophoresis using agarose (1.5% *w/v*) in 1× TAE buffer for 2 h at 70 V.

#### 4.3.3. DNA Sequencing of *fexA*

The amplified *fexA* product was subjected to Sanger sequencing commercially (Genetrics, Sciences, Dubai, United Arab Emirates). The obtained nucleotide sequences were compared with published fexA sequences, available online using the BLAST software available at the National Center for Biotechnology Information website: https://blast.ncbi.nlm.nih.gov/Blast.cgi (accessed on 14 June 2021). The DNA sequence was deposited in Genbank with the accession number Mz382798.

#### 4.3.4. Spa Typing

All isolates were *spa* typed as described by Harmsen et al. [45]. The PCR protocol consisted of an initial denaturation at 94 °C for 4 min, followed by 25 cycles of denaturation at 94 °C for 1 min, annealing at 56 °C for 1 min and extension for 3 min at 72 °C, and a final cycle with a single extension for 5 min at 72 °C. Five μL of the PCR product was analyzed by 1.5% agarose gel electrophoresis to confirm amplification. The amplified PCR product was purified using a Micro Elute Cycle-Pure Spin kit (Omega Bio-tek, Inc., Liburn, Georgia, USA), and the purified DNA was then used for sequencing PCR. The sequencing PCR product was then purified using an Ultra-Sep Dye Terminator Removal kit (Omega Bio-tek, Inc., Liburn, GA, USA). The purified DNA was sequenced in an automated 3130 × 1 genetic analyzer (Applied Biosystems, Waltham, MA, USA). The sequenced *spa* gene was analyzed using the Ridom Staph Type software (Ridom GmbH, Wurzburg, Germany).

#### 4.3.5. DNA Microarray

DNA microarray analysis was performed using the Identibac *S. aureus* genotyping kit 2.0 and the ArrayMate reader (Alere Technology, Jena, Germany) as described previously by Monecke et al. [46]. The DNA microarray analysis was used for the simultaneous detection of SCC*mec* types, antibiotic resistance genotypes, and virulence-related genes, including PVL, genes encoding species markers, and to allocate clonal complex (CC). *S. aureus* genotyping array is presented in an ArrayStrip format which contains 336 probes printed onto an array located in the bottom of the ArrayStrip. MRSA isolates were grown on blood agar plates at 35 °C overnight. DNA extraction of the overnight culture was performed as described by the manufacturer using Identibac *S. aureus* genotyping kit 2.0 (Alere, GmbH, Germany). Linear amplification of the purified DNA was performed in a total of 10 μL of the reaction volume containing 4.9 μL of B1 (labeling reagent), 0.1 μL of B2 (DNA polymerase), and 5 μL of the purified DNA. The PCR protocol consisted of an initial denaturation for 5 min at 96 °C, followed by 50 cycles of denaturation for 60 s at 96 °C, annealing for 20 s at 50 °C, and extension for 40 s at 72 °C. Hybridization and washing of the labelled arrays were performed as previously described [46]. The array was scanned using the ArrayMate reader (CLONDIAG, Alere, Germany), and the image of the arrays was recorded and analyzed using IconoClust software plug-in (CLONDIAG). The result was interpreted as negative, positive, or ambiguous by the software.

#### 4.3.6. Multilocus Sequencing Typing (MLST)

MLST was performed for representative isolates belonging to different *spa* types. The amplification of the seven housekeeping genes was performed using previously described M13-tailed primers [47]. The amplified targets were sequenced with one pair of M13-tailed primers: 5′- TGT AAA ACG ACG GCC AGT-3′ and 5′- CAG GAA ACA GCT ATG ACC-3′. The sequencing PCR protocol consisted of initial denaturation for 1 min at 94 °C, followed by 25 cycles of denaturation for 10 s at 96 °C, annealing at 55 °C for 5 s, and extension for 4 min at 66 °C. DNA sequencing was performed using a 3130 ȕ 1 Genetic analyzer (Applied Biosystems, Foster City, CA, USA) in accordance with the manufacturer’s protocol. The sequences were submitted to http://www.pubmlst.net/ where an allelic profile was generated, and the sequence type (ST) assigned.

### 4.4. Genetic Location of Chloramphenicol Resistance Determinants

#### 4.4.1. Plasmid Analysis

The isolation of plasmid DNA, curing and DNA transfer experiments by mixed-culture transfer, conjugation, and mobilization were performed as described previously [48]. Mobilization of non-conjugative plasmids was performed using conjugative plasmids pWBG636 (Gm^R^) and pXU12 (mup^R^) as described previously by [49]. Loss of resistance and plasmids by curing was performed by growing the organisms at 43 °C overnight and screening single colonies for the loss of resistance by a replica plating method. Colonies that lost antimicrobial resistance were screened for plasmid loss by agarose gel electrophoresis.

#### 4.4.2. Mixed Culture and Conjugation

For mixed-culture transfer (MCT), *S. aureus* strain WBG1876 was used as recipient [48]. Briefly, 0.2 mL volumes of overnight growth of donor and recipients were added to 5.0 mL of brain heart infusion broth (BHIB) containing 0.04 M CaCl_2_. The mixture was incubated overnight at 37 °C with gentle shaking, and cells were collected by centrifugation. The deposit was spread on brain heart infusion agar containing rifampicin (2.5 mg/L) together with one of the following agents (mg/L), mupirocin (10), tetracycline (5), and cadmium acetate (135). Selection plates were incubated for up to 48 h at 37 °C. Transcipients were screened for co-transfer of unselected resistance determinants and analyzed by agarose gel electrophoresis for plasmid contents.

For conjugation experiments, 2 mL volumes of overnight cultures of the donor and recipient (XU21) cells were mixed in a tube and centrifuged at 2000× *g* for 5 min. *S. aureus* strain XU21 is *S. aureus* strain RN4220 mutated to chromosomal resistance to novobiocin and rifampicin and was used as recipient in conjugation experiments [48] The deposit was resuspended in 0.5 mL of BHIB, and 5 mL of polyethylene 6000 (40% *w/v*) was added. The tubes were incubated overnight at 37 °C with gentle shaking (150 rpm). Following centrifugation at 5000 rpm for 5 min, the supernatants were discarded, and the deposit was resuspended in 1 mL BHIB and vortexed. Serial 10-fold dilutions were made, and 0.1 mL of each dilution was spread on to selection plates as in MCT. In both MCT and conjugation experiments, controls consisting of donor and recipient cell only were used. Transfer was positive when growth was obtained on selection plates from the donor plus recipient mixtures and not on selection media plated with either donor or recipient alone.

#### 4.4.3. Mobilisation

Mobilization consisted of two rounds of conjugation experiment. In the first round, the strain carrying the conjugative plasmid, pWBG636, (Gm^r^) was conjugated with the clinical isolate as the recipient strain (e.g., #14071) with transconjugants obtained on selection plates containing gentamicin (10 mg/L) and mupirocin (10 mg/L) or tetracycline (5 mg/L). When the conjugative plasmid was transferred to the recipient, the resultant strain was then used as donor in a second round of conjugation experiment with strain XU21 as recipient. Transconjugants were obtained on selection plates containing novobiocin (5 mg/L), rifampicin (2.5 mg/L) and one of mupirocin (10 mg/L), chloramphenicol (30 mg/L), erythromycin (15 mg/L), and tetracycline (5 mg/L). Transconjugants were screened for plasmid contents and resistance transferred.

## Figures and Tables

**Figure 1 antibiotics-10-01250-f001:**
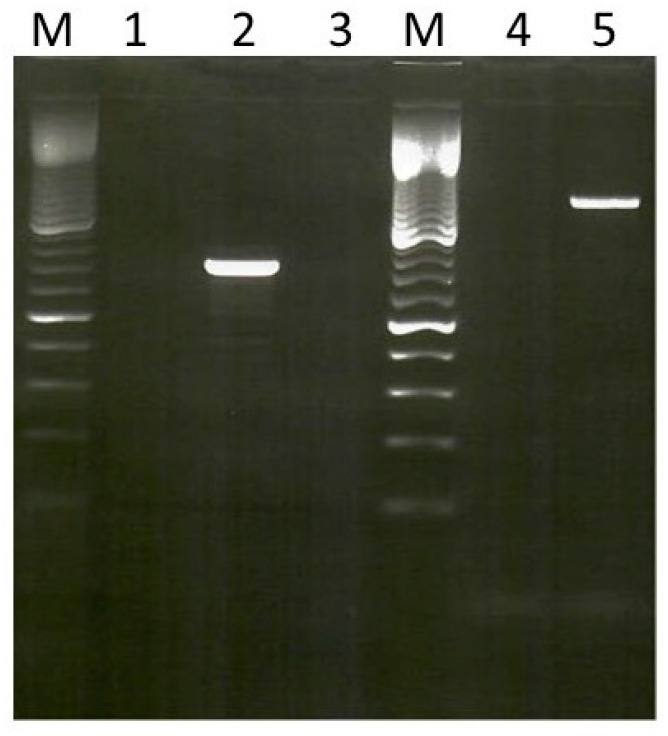
Amplification of chloramphenicol resistance genes in MRSA. Lanes M, 100 bp ladder molecular size markers; Lane 2, *cat*, (748 kb), Lane 5, *fexA* (1272 kb).

**Figure 2 antibiotics-10-01250-f002:**
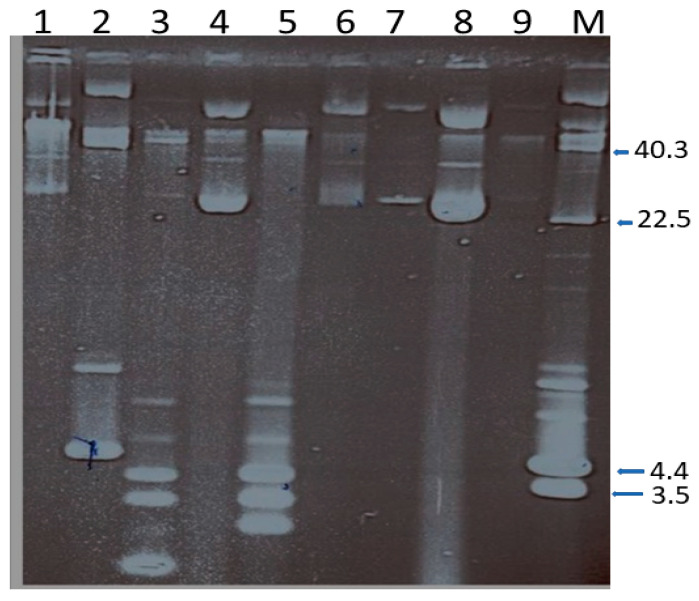
Plasmid content of chloramphenicol-resistant MRSA. Lanes 1, 4, 6,7,8, and 9 harbor single plasmids of c.28.0 kb in size. Lane 2 harbors two plasmids of c.40.0 and 4.4 kb; Lane 3 harbors four plasmid of c, 28.o, 3.5, 2.8 and <2.0 kb; Lane 5 harbors three plasmids of 3.5, 2.8 and 2.0 kb. Lane M contain plasmid size markers. The sizes are in kb. Only the CCC forms of the plasmids are labeled.

**Figure 3 antibiotics-10-01250-f003:**
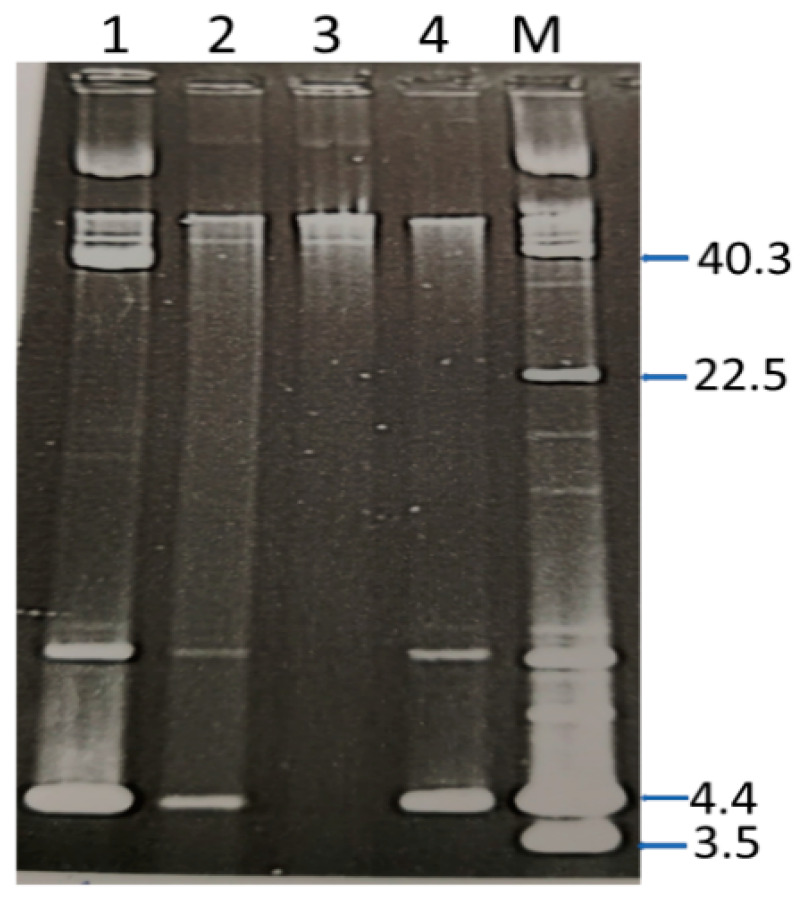
Transfer of gentamicin, mupirocin, and chloramphenicol resistance. Lane 1 Isolate #14282 harboring c.40 and 4.4 kb plasmids and resistant to chloramphenicol, mupirocin, and gentamicin. Lane 2, #14282 cured of c.40 kb plasmid and resistance to gentamycin and mupirocin; Lane 3, #14282 cured of c.40 kb and 4.4 kb plasmids and resistance to gentamicin, mupirocin, and chloramphenicol; Lane 4, Transconjugants of #14282 resistant to chloramphenicol. Lane M. Contain plasmid molecular size markers. Sizes are in kb. Only CCC forms of plasmid are labeled.

**Table 1 antibiotics-10-01250-t001:** Characteristics of chloramphenicol-resistant MRSA isolates.

S/N	Strain Description	ST	Spa Type	#	ArcC	AroE	GlpF	GmK	ptA	tpi	YqiL
1	CC5-MRSA-VI+SCCfus	627	t688	43	1	4	1	4	1	1	10
2	CC5-MRSA-VI+SCCfus	627	t450	1	1	4	1	4	1	1	10
3	CC5-MRSA-VI+SCCfus	627	t954	1	1	4	1	4	1	1	10
4	CC5.MRSA-V	5	t688	6	1	4	1	4	2	1	3
5	CC8-MRSA-III	239	t037	2	2	3	1	1	4	4	3
6	CC8-MRSA-III	239	t860	1	2	3	1	1	4	4	3

**Abbreviations:** ArcC (Carbamate kinase); AroE (Shikimate dehydrogenase); GlpF (Glycerol kinase); Gmk (Guanylate kinase); ptA (Phosphate acetyltransferase); tpi (Triosephosphate isomerase); YqiL (Acetyl coenzyme A acetyltransferase), #, number of isolates.

**Table 2 antibiotics-10-01250-t002:** Antibiotic resistance of chloramphenicol resistant MRSA isolates.

S/N	MRSA Clones	#	Antibiotic Resistance	Resistance Genes
**1**	ST5-V-t688 (WA MRSA 11/34)	5	Em, Clin, Tet, Cip	*fexA, erm(C), tet(K), tet(M)*
**2**	ST5-V-t688 (WA MRSA 81/85)	1	Em, Clin, Tet	*fexA, erm(C), tet(K)*
**3**	ST627-VI-t450 (MRSA-VI +SCC*fus*)	1	Tet, Tp, Fd	*fexA, fusC, dfrS1, tet(M)*
**4**	ST627-VI-t688 (MRSA-VI +SCC*fus*)	43	Tet, Tp, Fd	*fexA, fusC, dfrS1, tet(M)*
**5**	ST627-VI-t954 (MRSA-VI +SCC*fus*)	1	Tet, Tp, Fd	*fexA, fusC, dfrS1, tet(M)*
**6**	ST239-III-t037 (Vienna/Brazilian)	2	Gm, Km, Em, Clin, Tet, Tp, Fd	*cat, aacA-aphD, aphA3, tet(K), tet(M), erm(A)*
**7**	ST239-III-t860 (Vienna/Brazilian)	1	Gm, Km, Em, Clin, Tet, Tp, Fd, Mup	*cat, aacA-aphD, aadD, aphA3, tet(M), erm(A), mupA*

**Abbreviations:** Clin, clindamycin; Cip, ciprofloxacin; Em, erythromycin; Fd, fusidic acid; Gm, gentamicin; Km, kanamycin; Mup, mupirocin; Tet, tetracycline; Tp, trimethoprim; *aacA-aphD*, aminoglycoside adenyl-/phoshotransferase; *aadD*, aminoglycoside adenyl transferase; *aphA3*, aminoglycoside phosphotransferase; *cat*, chloramphenicol acetyl transferase; *dfrS1*, dihydrofolate reductase mediating trimethoprim resistance; *erm(A)*; rRNA methyltransferase (A), *erm(C)*, rRNA methyltransferase (C); *fexA*, chloramphenicol/florfenicol exporter; *fusC*, fusidic acid resistance gene (Q6GD50); *mupA*, isoleucyl-tRNA synthethase associated with mupirocin resistance; *tet(K)*, tetracycline efflux protein; *tet(M)*, ribosomal protection protein associated with tetracycline resistance’ #, number of isolates.

**Table 3 antibiotics-10-01250-t003:** Loss of resistance and plasmids in Chloramphenicol-resistant MRSA isolates.

S/N	MRSA Strain	Resistance Profile	Plasmid Content, kb	Resistance Lost	Plasmids Lost, kb
1	13973 (ST5-V-t688)	Cm, Em, Clin, Tet	28.0, 2.8	Em	2.8
2	14071 (ST5-V-t688)	Cm, Em, Clin, Tet	c.40.0, 28.0	None	None
3	14098 (ST627-VI-t688)	Cm, Tet, Tp, Fd	28.0	None	None
4	14284 (ST239-III-t037)	Cm, Gm, Km, Em, Clin, Cip, Fd, Mup	40.0, 4.4	Cm, Gm, Km, Mup	4.4, 40.0
5	14299 (ST239-III-037)	Cm, Gm, Km, Em, Clin, Tet, Fd	40.0, 3.5, 2.8, <2.0	Cm	3.5
6	14314 (ST5-V-t688)	Cm, Tet, Tp, Fd	28.0	None	None
7	14387 (ST239-III-t860)	Cm, Gm, Km, Em, Clin, Tet, Fd	28.0, 3.5, 2.8, 2.0	Cm, Em, Clin	3.5, 2.8
8	14434 (ST627-VI-t688)	Cm, Tet, Tp, Fd	28.0	None	None

**Abbreviations.** Clin, clindamycin; Cip, ciprofloxacin; Em, erythromycin; Fd, fusidic acid; Gm, gentamicin; Km, kanamycin; Mup, mupirocin; Tet, tetracycline; Tp, trimethoprim.

**Table 4 antibiotics-10-01250-t004:** Transfer of chloramphenicol resistance.

Isolates	Resistance Profile	Plasmid Content kb	* Mode of Transfer	Resistance Transferred	Plasmid Transferred, kb	Resistance Genes
**14284** **ST239-III-t037**	Cm, Gm, Km, Em, Clin, Cip, Fd, Mup	c.40.0, 4.4	C	Gm, Mup, Cm	c.40.0, 4.4	*cat, mupA, aacA-aphD*
			C	Gm, Mup	c.40.0	*mupA, aacA-aphD*
			C	Cm	4.0	*cat*
**14299** **ST239-III-t037**	Cm, Gm, Km, Em, Clin, Tet, Tp, Fd	c.40, 4.4	M	Cm	3.5	*cat*
**14387** **ST239-III-t860**	Cm, Gm, Km, Em, Clin, Tet, Fd	28.0, 3.5, 2.8 2.0	M	Cm	3.5	*cat*
**13973** **ST5-V-t688**	Cm, Em, Clin, Tet	28.0, 2.8	M	Em	2.8	*erm(C)*
**14434** **ST627-VI-t688**	Cm, Tet, Tp, Fd	28.0	C, M	None	None	None
**14071** **ST5-V-t688**	Cm, Em, Clin, Tet	c.40.0, 28.0	C, M	None	None	None
**14098** **ST627-VI-t688**	Cm, Tet, Tp, Fd	28.0	C, M	None	None	None

**Abbreviations:** Kb, kilobase; * C, conjugation; * M, mobilization; Cm, chloramphenicol; Em, erythromycin; Clin, clindamycin; Tet, tetracycline; Gm, gentamicin; Km, kanamycin; Tp, trimethoprim, Cip, ciprofloxacin; Fd, fusidic acid.

## Data Availability

Data are available within the article.

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
