# Peer review of "Resurgence of Chloramphenicol Resistance in Methicillin-Resistant Staphylococcus aureus Due to the Acquisition of a Variant Florfenicol Exporter (fexAv)-Mediated Chloramphenicol Resistance in Kuwait Hospitals"

_antibiotics, 2021, doi:10.3390/antibiotics10101250_

Round 1

Reviewer 1 Report

The manuscript “Resurgence of chloramphenicol resistance in methicillin-resistant staphylococcus aureus due to the acquisition of a variant 2 florfenicol exporter (fexAv)-mediated chloramphenicol resistance in Kuwait hospitals” by Udo et al. referred to global problem of antibiotic resistance, especially resistance to methicillin of S. aureus (MRSA). Resistance to chloramphenicol of MRSA strains is a worldwide poorly understood question. The Authors showed that the surge in the proportion of chloramphenicol-resistant MRSA isolates was due to the introduction and spread of closely related novel MRSA clones belonging to ST627-VI-t688 and ST5-V-t688 in Kuwait hospitals. The clones carried a variant fexA that mediated resistance to chloramphenicol but not florfenicol. The manuscript is well-written and is well-presented. There are an interesting idea, labour-intensive and multiple experiments, and fructuous results.

However, some data need to be changed.

  1. In my opinion, the title is too long and not very communicative. It should be modified.
  2. The Abstract is too long, over 200 words.
  3. Why were strains isolated six years ago used in the paper? Did the Authors not have newer material for the study? I wonder what the situation is now?
  4. Table 3 should be completed. There are only “+” in the What do the blanks mean? “-“ or maybe not tested?
  5. Figure 2 and Figure 3 (photos) are not of good quality, maybe they can be improved somehow.
  6. Some typos in the manuscript should be corrected, e.g. gene names not in italics.

Author Response

Title: Resurgence of Chloramphenicol resistance in Methicillin-Resistant Staphylococcus aureus due to the acquisition of a variant Florfenicol exporter (fexAv)-mediated Chloramphenicol resistance in Kuwait hospitals. 

Reviewer #1.

  1. Comments

In my opinion, the title is too long and not very communicative. It could be modified.

Response

 We are grateful to this reviewer for his/her time and input to improve our paper. However, regarding the title, although long, we feel that it confers the message embodied in the paper. As a result, we prefer to keep it as it is.

  1. Comments

The abstract is too long, over 200 words.

Response

We have checked the word count of the abstract. It is less than 200 words (188 words).

  1. Comments

Why were strains isolated six years ago used in this paper? Did the Authors not have newer material for the study?  I wonder what the situation is now?.

               Response

  We have newer isolates. However, we started working on these isolates in 2016.  Unfortunately, it took some time to get to the point when we could submit the manuscript for publication.   The fexA -positive strains have continued to be isolated in the state. We have added a statement to this effect in the conclusions on page 20, lines 404-406.

  1. Comment

Table 3 should be completed. There are only “+” in the What do the blanks mean

? “-“  or maybe not tested 

Response

The blank have been filled with “-“meaning not detected. Table 3 has been moved to a supplementary file designated Table S1 on the recommendation of the other reviewers.

  1. Comment

Figure 2 and Figure 3 (photos) are not of good quality, maybe they can be improved somehow

Response.

We have added new figures with improved quality.

  1. Comments

Some typos in the manuscript should be corrected, e.g. gene names not in italics.

Response

We have made the necessary corrections.

Reviewer 2 Report

This manuscript describes the clinical isolated Staphylococcus aureus MRSA strains that express the CMR  phenotypes but susceptible to florfenicol.  Among them, the fexA Av (florfenicol exporter varients)-positive isolates mostly belonged to  CC5-ST627-VI clones (n=45) and  CC5-ST5-V-t688 (n=6).  The fexA variants that confer chloramphenicol but not florfenicol resistance. The cat-positives (CMR) isolates were CC8-ST239-III (n=3).  While cat gene was carried on 3.5~4.4 kb plasmids, the location of fexA gene could not be localized. The resurgence of chloramphenicol resistance was due to the closely related fexA-positive CC5-ST5-V and CC5-ST627-VI clones.

Suggestions:

Major concern:

(1) In Figure 1. Lanes M, 100bp molecular size markers can changed to 100bp ladder molecular size markers.

(2) The table 3 could put  in the supplementary data, since all the data derived from the microarray analyses. However the 2.6 . virulence genes of chloramphenicol resistant isolates could still be there.

(3) Since the fexA variants (fexAV) could be due to the evolutionary alterations and the authors did not discuss this possibility and any functional change for this mutant as compared to the wild-type fexAV.  The directed evolutionary mechanisms could be also considered.  As usually, the natural selection mutants could match the chemical treatment induced mutations.

Minor suggestions:

(1) line 24: fexA- positive  changed to fexA-positive

(2) line 135 and 136 should be connected well, there is not needed space region 

(3) line 167 All, isolates  changed to All isolates

(4) line 278 in this study.. should be in this study.

(5) line 287 with he  should be with the

(6) line 293 a single patient in 2010 has successfully     add a comma after 2010

(7) line 363 swabs (n=8) blood   add a comma after (n=8) 

(8) line 378 as quality control  changed as the positive control 

(9) line 476 with gentle shaking. (150rpm)  changed to gentle shaking (150rpm).

(10) line 474 recipient (XU21)  please describe the the bacteria (XU21) 

(11) line 477 at 2000g for 5 min the supernatants changed to 2000g for 5 min, the supernantants (another suggestions 2000g can be changed to rpm)

Author Response

Reviewer # 2.

  1. Comment

In Figure 1, lane M, 100 bp molecular markers can be changed to 100bp ladder molecular size markers. 

Response

This has been done. Please see Page 9, line 181.

  1. Comment

Table 3 could be put in the supplementary data, since all the data derived from the microarray analyses. However, the 2.6 virulence genes for chloramphenicol resistant isolates could still be there.

Response

We have moved Table 3 to supplementary data designated Table S1. We have retained section 2.6 in the manuscript. Please see page 10. 

Since the fexA variant (fexAv) could be due to the evolutionary alterations and the authors did not discuss this possibility and any functional change for this mutant as compared to wild-type fexAv. The directed evolutionary mechanism could also be considered. As usually, the natural selection mutant could match the chemical treatment induced mutations.

Response.

We appreciate this insightful comment.  However, we feel that we do not have sufficient data to address the points raised at this time.

Minor suggestions.

  1. Line 24, fexA- positive changed to fexA-positive

Response

This has been done. Line 40.

  1. Line 135 and 136 should be connected well, there is not needed a space region.

Response

This has been done. Line 160-161.

  1. Line 167 All, isolates changed to All isolates.

Response.

This has been done. Line 199.

  1. Line 278 in this study.. should be in this study.

Response

This has been changed. Line 330.

  1. Line 287 with he should be with the.

Response

This has been changed. Line 339

  1. Line 293 a single patient in 2010 has successfully add a coma after 2010.

Response

This has been done. Line 345.

  1. Line 363 swabs (n=8) blood add a coma after (n=8).

Response

This has been done. Line 416

  1. Line 378 as quality control changed as positive control.

Response

We prefer to retain “quality control” since the test result is neither positive or negative but resistant or susceptible.

  1. Line 476 with gentle shaking. (150 rpm) changed to gentle shaking (150 rpm).

Response

This has been changed. Line 525

  1. Line 474 recipient (XU21) please describe the bacteria (XU21).

Response

A description of XU21 has been provided on Line 521-523.

  1. Line 477 at 2000g for 5 min the supernatants  changed to 2000g for 5 min,  the supernatant (another suggestions 2000g can be changed to rpm)

Response

“ for 5 min the supernatant” was changed to “ for 5 min, the supernatant”.  2000g has been converted to 5000 rpm.  Line 525.

Reviewer 3 Report

Udo et al., have studied in this work the chloramphenicol resistance in some MSRA isolates in Kuwait hospitals. The experiments are well designed and executed but the interest on the subject is very low, because chloramphenicol is not used any more and nobody care about this antibiotic resistance. On the other hand the result could be interest as a report about antibiotic resistance.

Additionally, there are serious mistakes in the introduction (chloramphenicol irreversibly bound) or  missing information about chloramphenicol mode of action.

I suggest to be rejected in the current form and resubmitted in a new with shorted extent  (without table 3 and long discussion) and corrected introduction.

Many data and experimental details have to be moved in supplementary data.

Author Response

Reviewer #3

Comments and suggestions.

  1. Udo et al., have studied in this work the chloramphenicol resistance in some MRSA isolates in Kuwait hospitals. The experiments are well designed and executed but the interest on the subject is very low, because chloramphenicol is not used anymore and nobody care about this antibiotic resistance. On the other hand the result could be of interest as a report about antibiotic resistance

Response

We thank the reviewer for appreciating the quality of the study design and execution. We agree that chloramphenicol is no longer widely used therapeutically in many parts of the world including Kuwait.  We were surprised to see the high numbers of isolates that expressed resistance to chloramphenicol when the antibiotic was not used for treatment. This is what necessitated this study in the first place. In this case, chloramphenicol resistance served as a marker to inform on the introduction and local transmission of novel MRSA clones which is the main message of this paper. The second important message is the fact that this clone, carrying a mutant fexA determinant similar to the isolates obtained from chicken meat, may have been acquired from chicken. Thirdly, the chloramphenicol-resistant strains carried varied virulence factors and caused different types of infections; points that cannot be ignored.

  1. Additionally, there are serious mistakes in the introduction (chloramphenicol irreversibly bound) or missing information about chloramphenicol mode of action.

Response.

We thank the reviewer for drawing our attention to this error. We have deleted “irreversibly” from the text. Please see lines 60-61

  1. I suggest to be rejected in the current form and resubmitted in new with shorted extent (without table 3 and long discussion and corrected introduction.

Many data and experimental details have to be moved in supplementary data.

Response

Table 3 has been removed from the text and placed as a supplementary table, Table S1. The error in the introduction has been corrected.

Round 2

Reviewer 3 Report

The introduction must be improved. References are missing, ie

Antonelli et al., 2018

Svetlov et al, 2019....

Author Response

  1. Comments

The introduction must be improved. References are missing, ie

Antonelli et al., 2018

Svetlov et al., 2019.

Response

 We are grateful to this reviewer for drawing out attention to these two references that were not used previously in the paper.

We have used these papers as references #6 and #40 in the revised manuscript.  A paragraph in relation to ref.#40 has been added to the Discussion n Lines 398-405.

We have modified the sentence in Line 60 of the introduction.

This manuscript is a resubmission of an earlier submission. The following is a list of the peer review reports and author responses from that submission.

Round 1

Reviewer 1 Report

This manuscript, entitled “Resurgence of hloramphenicol resistance in Methicillin-Resistant Staphylococcus aureus due to the acquisition of a variant Florfenicol exporter (fexAv)-mediated Chloramphenicol resistance in Kuwait hospitals”, had described the genotypes and antibiotic resistance of the chloramphenicol-resistant MRSA isolates to ascertain in Kuwait hospitals. The study on clinical isolates is of importance, however, the information and novelty provided in this manuscript are limited, thus, my suggestion is rejection.

Most importantly, the isolates are collected >5 years ago, which lesser the novelty and significance. Also, the sample number is limited.

Secondly, I recommend the authors to perform genome sequencing on the clinical isolates to show more information concerning the carriage of antimicrobial resistance determinants and virulence gene, and correspond to their phenotypes.

The gels in the figures are too vague, making the result less convincing.

Reviewer 2 Report

This study was initiated to explain the possible causes of the resurgence in chloramphenicol- resistant MRSA in Kuwait hospitals as the resurgence in the prevalence of chloramphenicol resistance in Kuwait MRSA isolates could not be explained by increased chloramphenicol use because chloramphenicol is hardly used for treatment in this country.

Molecular typing revealed that most of the chloramphenicol-resistant isolates belonged to CC5-ST627-VI-t688/t450/t954 (45/54) or CC5-ST5-V-t688 (6/54) while a small number (3/54) belonged to ST239-III/t037/t860. This indicates clearly that the surge in the proportion of chloramphenicol-resistant MRSA isolates was due to the introduction and spread of closely related novel MRSA clones belonging to ST627-VI-t688 and ST5-V-t688.

The results of this study suggests that the CC5-MRSA-VI+SCCfus clone, carrying the unique composite genetic element consisting of SCCmec VI and fusidic acid resistance determinant, has successfully spread to become the dominant chloramphenicol-resistant clone in Kuwait hospitals.

This study further revealed that two chloramphenicol resistance determinants namely, cat, that codes for chloramphenicol acetyltransferase and fexA that codes for chloramphenicol/florfenicol exporter conferred chloramphenicol resistance in Kuwait.

Although fexA usually confers combined resistance to chloramphenicol and florfenicol in S. aureus obtained from animals and humans, the fexA- positive isolates in this study were resistant to chloramphenicol but susceptible to florfenicol

A comparison of the DNA sequence of the fexA in the isolates in this study yielded 100% similarity to the variant form of fexA (fexAv) reported in MRSA, including mutations leading to amino acid substitutions.

Ιn conclusion, this study revealed that the resurgence in chloramphenicol resistance in MRSA obtained in Kuwait hospital was due to the introduction of a new chloramphenicol-resistant MRSA clone harboring a variant fexA that mediated resistance to chloramphenicol but not florfenicol. The variant fexA shared 100% sequence similarity with a fexA variant that was detected in MRSA isolated from chicken meat. The similarity of these isolates to those obtained from chicken meat and milk from cattle and buffalos points to the ongoing intrusion of MRSA isolates from livestock to humans.

I think that this is an interesting study offering a large amount of information regarding Chlroramphenicol resistance in Kuwait despite its limited use and I suggest that the paper is suitable for publication in its current format.